# Age-Related Decline of Male Fertility: Mitochondrial Dysfunction and the Antioxidant Interventions

**DOI:** 10.3390/ph15050519

**Published:** 2022-04-23

**Authors:** Jing-Jing Wang, Shu-Xia Wang, Yan Feng, Rui-Fen Zhang, Xin-Yue Li, Qiong Sun, Jian Ding

**Affiliations:** School of Life Science and Technology, Xi’an Jiaotong University, Xi’an 710049, China; 18681881702@163.com (J.-J.W.); sxwang13@163.com (S.-X.W.); tehminakhancom@gmail.com (T.); 17765080352@163.com (Y.F.); ruifenzhang1996@163.com (R.-F.Z.); l9898xy@stu.xjtu.edu.cn (X.-Y.L.)

**Keywords:** mitochondria, reactive oxidation species, aging, antioxidant, male infertility

## Abstract

Mitochondria are structurally and functionally unique organelles in male gametes. Apparently, as the only organelles remaining in mature sperm, mitochondria not only produce adeno-sine triphosphate (ATP) through oxidative phosphorylation (OXPHOS) to support sperm mobility, but also play key roles in regulating reactive oxidation species (ROS) signaling, calcium homeostasis, steroid hormone biosynthesis, and apoptosis. Mitochondrial dysfunction is often associated with the aging process. Age-dependent alterations of the epididymis can cause alterations in sperm mitochondrial functioning. The resultant cellular defects in sperm have been implicated in male infertility. Among these, oxidative stress (OS) due to the overproduction of ROS in mitochondria may represent one of the major causes of these disorders. Excessive ROS can trigger DNA damage, disturb calcium homeostasis, impair OXPHOS, disrupt the integrity of the sperm lipid membrane, and induce apoptosis. Given these facts, scavenging ROS by antioxidants hold great potential in terms of finding promising therapeutic strategies to treat male infertility. Here, we summarize the progress made in understanding mitochondrial dysfunction, aging, and male infertility. The clinical potential of antioxidant interventions was also discussed.

## 1. Introduction

The World Health Organization (WHO) has estimated that approximately 190 million people worldwide suffer from infertility, which has become the third most commonly diagnosed disease after cancer and cardiovascular/cerebrovascular disorders [1]. Currently, the incidence of infertility in couples of childbearing age globally is as high as 10–15%, and this number is increasing year by year. Approximately 50% of cases of infertility are partially or wholly attributable to male factors [2,3]. Given the prevalence and underlying complexity of male infertility, a better understanding of sperm pathophysiology is very important and is expected to offer more insights into the treatment of this condition.

Normal sperm structure and function underlie normal male fertility [4]. A mature sperm is a highly specialized cell with a unique structure consisting of a head and a tail (or flagellum) (Figure 1). The head contains a condensed haploid nucleus in a very small amount of cytoplasm. Unlike other types of cells, sperm are unencumbered by typical cytoplasmic organelles, such as the endoplasmic reticulum or Golgi apparatus. Notably, mitochondria are the major endomembrane organelles in the flagella of sperm. Mitochondria not only play a central role in cellular energy production, but also participate in the regulation of redox and calcium homeostasis, steroid hormone biosynthesis, and apoptosis [5,6]. However, the underlying processes are complex and remain incompletely understood [7,8].

Each sperm contains ~70 mitochondria. The organelle usually has a condensed morphology and compacted matrix which allow for highly efficient oxidative phosphorylation (OXPHOS) and energy production. Electron leakage in the electron transport chain (ETC) also produces reactive oxygen species (ROS) as a by-product. ROS derived from sperm mitochondria are a “double-edged sword”. ROS at the physiological level are essential for spermatogenesis, sperm capacitation, and fertilization, while an excess of ROS can cause oxidative stress (OS), lead to sperm lipid peroxidation and DNA damage, and result in male infertility [9,10]. There are many causes of male infertility, including genetic, endocrine, lifestyle, and/or age-related factors. Some infections can interfere with sperm health or even block the passage of sperm. Varicoceles, ejaculation issues, and defects of the tubules transporting sperm cause male infertility, too. Notably, many of the cases of male infertility involve OS. Approximately 50% of male infertility is attributed to the presence of OS in the reproduction system.

Male fertility declines concurrent with the aging process. The negative impacts of aging on semen parameters, including semen volume, sperm count, motility, and morphology have been demonstrated in numerous studies [11]. The aging of the male reproductive system involves multi-factorial changes. Alterations to mitochondria are among the most remarkable features contributing to lower fertility. In particular, the accumulation of oxidative damage resulting from mitochondrial defects can impair the normal function of reproductive cells and lead to the loss of male fertility [12].

Seminal fluid is mainly composed of sperm and seminal plasma, the latter being the main part of semen, representing more than 90% of its volume. Seminal plasma is the living environment of sperm, in addition to containing large amounts of water, inorganic ions, carbohydrates (e.g., glucose, fructose), lipids (e.g., cholesterol), proteins, and polypeptides, there are many cytokines (e.g., interferon-γ, tumor necrosis factor-α, interleukin-2, interleukin-4), specific hormones (e.g., testosterone, anti-Mullerian hormone, luteinizing hormone, and follicle-stimulating hormone), and enzymes (e.g., neutral α-glucosidase), which can provide energy and nutrients for sperm activity [13,14]. Sperm possess a limited supply of antioxidant enzymes due to the minimal availability of cytoplasmic space. Fortunately, the enzymatic antioxidants (e.g., superoxide dismutase and catalase) and non-enzymatic antioxidants (e.g., vitamins C and E, glutathione, carnitine, and carotenoids) present in the seminal plasma play important protective roles [6,15,16]. Antioxidants are “scavengers” of ROS, which help maintain a dynamic oxidant/antioxidant balance in the body [17]. Several studies have found that antioxidants in the seminal plasma of male infertile patients are significantly decreased while ROS levels are increased [18,19]. Therefore, exogenous antioxidant supplementation may help improve semen quality. Several clinical studies have reported that oral antioxidants can help reduce OS and improve male fertility [20,21]. Antioxidant interventions could have beneficial effects and mitigate the age-related loss of male fertility. However, it is still necessary to evaluate the long-term effects in various settings. In this review, we aimed to summarize the existing literature on sperm mitochondria and the progress made in understanding mitochondria in aging and in the pathogenesis of male infertility, thereby providing new ideas for the diagnosis and treatment of this condition.

## 2. Mitochondria in Spermatozoa

Sperm are the mature germ cells of mammalian males. In the testis, sperm go through several stages of development, namely spermatogonia, primary spermatocytes, secondary spermatocytes, spermatids, and sperm. Mitochondria are present during all stages of spermatogenesis and sperm development with their morphology and distribution undergoing a series of dynamic changes [22]. In the Golgi, headgear, and early acrosome stages of spermatogenesis, mitochondria are concentrated and distributed throughout the cell. At these stages, the mitochondria are flat, small, round or oval, and their number increases significantly. In the late acrosome and early maturation stages of spermatogenesis, a portion of the organelles migrate to the flagellum, while the remainder aggregate and enter the remnant, where they are removed by Sertoli cells or through the autolysis of mature sperm cells [23]. During the late maturation stage, mitochondria begin to shrink and elongate, become crescent-shaped on the dense fibers that surround the axoneme in the middle of the sperm, and then transform into a rod shape, following which they line up end to end and wind around the middle part of the spermatozoa tail, forming a mitochondrial sheath (Figure 1) [24]. Thus, mitochondrial fusion and fission (mitochondrial dynamics) realize the dynamic changes in mitochondrial number and structure during spermatogenesis, allowing fine-tuned responses to cellular energy demands [6,25]. In addition, through mitochondrial dynamics, mitochondria in mature sperm are highly concentrated, and the mitochondrial metabolism is more efficient at this time, which also plays a key role in maintaining sperm motility and fertilization [22,26]. However, the role of mitochondrial dynamics during spermatogenesis and sperm maturation remains largely unknown.

During spermatogenesis and sperm development, with the disappearance of other cytoplasm organelles, such as the endoplasmic reticulum or Golgi apparatus, the number of mitochondria also changes significantly. However, mitochondria apparently are the only organelles remaining in mature sperm. There are approximately 72–80 mitochondria in mature sperm, which are distributed in the midpiece of the sperm flagellum and arranged in a spiral shape (10–12) to form a thick mitochondrial sheath (Figure 1) [24]. The mitochondrial sheath probably is one of the main sites of sperm metabolism and energy production.

Mitochondria are required for the normal function of sperm. Alterations of sperm mitochondrial structure, accompanied by changes in the content of various energy metabolism-related enzymes, such as cytochrome oxidase, succinate dehydrogenase, and lactate dehydrogenase, have been found to affect the energy supply of sperm and lead to sperm motility disorders. Studies have shown that the mid-flagellar structure can affect sperm motility and metabolism, and that changes in mitochondrial volume are often associated with altered ATP production and sperm speed [27]. These observations highlight the importance of the organelle in the maintenance of sperm motility.

## 3. Sperm Mitochondrial Dysfunction, Male Infertility, and Aging

Mitochondria are the key components of energy conversion and metabolism in sperm and participate in a variety of physiological processes such as spermatogenesis; sperm motility, hyperactivation, and capacitation; the acrosome reaction; and fertilization [8]. In addition to the production of ATP, mitochondria are also involved in various key cellular processes, including ROS production, calcium homeostasis, mitophagy, apoptosis, and steroid hormone biosynthesis [5,28].

### 3.1. Bioenergetic Roles of Mitochondria in Sperm and Male Fertility

ATP in sperm is exclusively produced via two metabolic pathways, namely glycolysis and OXPHOS. The former mainly occurs in the sperm head and the principal piece of the flagellum, while the latter primarily occurs in the region of mitochondrial distribution in the midpiece of the flagellum (Figure 1) [24]. However, which type of energy metabolism is preferentially used by sperm remains controversial. It is widely believed that glycolysis is more suited to sperm energy supply than the more complex OXPHOS because of the fewer links involved and the faster reaction speed. Additionally, owing to the mutual sliding process of the “9 + 2” microtubular structure of the sperm flagellum, a large amount of ATP is needed for sperm activity, and this requires a source of rapid ATP supply. Notably, the mitochondria are located in the midpiece of the sperm flagellum and far from the tip. Although, presumably, the ATP they generate cannot be rapidly transported to the axoneme tip [6]. The sperm of mice lacking the capacity for mitochondrial OXPHOS exhibited reduced fertilization ability [29]. These data indicated that ATP produced through OXPHOS still presents an important energy source needed for sperm motility.

The bioenergetic characteristics of mitochondria alter during aging. The respiratory activity and OXPHOS function correlated with the reproductive cycle, displaying a peak of functionality in young adult animals but declining in older animals [30,31]. These changes in testicular mitochondria can hinder ATP synthesis and consequently lead to an energy crisis which would affect the maintenance of testicular homeostasis [12].

### 3.2. Sperm ROS Content and Male Fertility

In mature sperm, mitochondria are a significant source of cellular ROS. This is mainly attributed to electron leakage in the ETC during the mitochondrial OXPHOS process (Figure 2). Electrons “escape” from multiple locations in the ETC, such as complex I and complex III. The escaped electrons inhibit the COX-mediated reduction of oxygen to water and react with O_2_ to form O_2_^−^; approximately 0.4–4% of the O_2_ consumed by mitochondria is converted to O_2_^−^ [10].

ROS are a “double-edged sword” in regulating sperm function [9]. Under physiological conditions, an appropriate concentration of ROS is necessary for sperm maturation in the epididymis [32,33]. In particular, O_2_^−^, can activate the cAMP/PKA pathway, which regulates the signal activity of various downstream pathways by promoting the phosphorylation of serine and tyrosine residues in target proteins and activating protein tyrosine kinases (PTKs). These signaling cascades can induce sperm capacitation and modulate sperm motility, thus significantly improving the ability of sperm to bind zona pellucida [34]. In addition, at physiological levels, ROS can increase membrane fluidity to promote sperm–egg fusion. During the capacitation process, ROS prevent the inactivation of phospholipase A2 (PLA2) by inhibiting protein tyrosine phosphatase activity. PLA2 can separate secondary fatty acids from membrane phospholipids, thereby increasing membrane fluidity [35]. In summary, physiological ROS levels play an important role in sperm proliferation and maturation, capacitation, hyperactivation, and acrosome reaction, as well as fertilization.

In contrast, excessive ROS can damage the sperm lipid membrane and DNA and impair sperm motility [17]. Sperm cell membranes are rich in a variety of unsaturated fatty acids, which renders them more vulnerable to ROS-induced lipid peroxidation. Interestingly, OS-induced lipid peroxidation has also been shown in human sperm to generate electrophilic aldehydes, such as 4-hydroxynonenal and acrolein. These compounds react with mitochondria by targeting succinate dehydrogenase, leading to changes in respiratory chain function and the activation of apoptotic pathways, all of which lead to the overproduction of ROS [6]. Additionally, excessive ROS production can trigger DNA damage in sperm, causing errors in transcription and translation, thereby affecting sperm motility. Numerous clinical reports have confirmed that ROS levels are high in the semen of 25–40% of infertile male patients [36]. ROS levels in semen can be used as a biomarker, which is important for the diagnosis of male infertility, especially unexplained male infertility.

The levels of superoxide anion and the resultant lipid peroxides in cells increase with age. This is also often accompanied with the reduced activity of antioxidant enzymes. Actually, the balance between pro- and antioxidative agents appears to be altered in aging testis mitochondria with a shift towards the pro-oxidizing condition [12,37]. These results indicate that the OS, mostly due to the overproduction of free radicals in mitochondria, may represent one of the major causes of age-related male fertility loss.

### 3.3. Sperm Mitochondrial Membrane Potential and Male Fertility

The mitochondrial ETC is composed of four multimeric complexes (I–IV). Complexes I, III, and IV pump protons from the matrix into the IMS, thus generating a difference in electric potential across the IMM, which is also known as the mitochondrial membrane potential (MMP, Δψm). The MMP represents an intermediate form of energy storage during ATP synthesis and can reflect the process of mitochondrial electron transmission and OXPHOS. A normal sperm MMP is the essential physiological basis for maintaining sperm acrosomal enzyme activity and chromatin integrity, and sperm with low MMP are not prone to acrosome reaction [8,38].

Numerous studies have confirmed that the MMP is positively correlated with sperm motility. For instance, the sperm of patients with severe asthenospermia exhibit an extremely low MMP [39]. Decreased MMP has also been detected in aging [40]. Recently, it has been proposed that MMP can serve as a marker to predict sperm fertilization ability in both natural conception and IVF, and is of great significance for the clinical evaluation of male fertility [41]. In particular, the combination of the MMP and sperm DNA fragmentation index (DFI) is even better than conventional semen parameters in predicting the success of natural conception in females [42] and is an important index for the clinical evaluation of idiopathic male infertility [43].

### 3.4. Calcium Homeostasis and Male Fertility

The calcium is one of the most abundant elements in mammals. Altered calcium homeostasis is generally associated with aging. Ca^2+^, acting primarily as a second messenger, plays a crucial role in the regulation of a wide range of cellular processes, including sperm motility, capacitation, hyperactivation, the acrosome reaction, and chemotaxis [6]. Mitochondria are important storage sites for intracellular Ca^2+^. The appropriate concentration of Ca^2+^ in the matrix can promote energy generation, thus increasing intracellular ATP content and improving sperm motility. Ca^2+^ absorption by mitochondria is mediated by different channels, especially the mitochondrial calcium uniporter (MCU). Blocking the MCU has been shown to reduce human sperm motility and ATP levels [44]. These results underline the importance of mitochondria in Ca^2+^ regulation and sperm function.

Mitochondria has been proposed to participate in regulating intracellular calcium signaling [45]. However, the roles sperm mitochondria play in this process is open to question, given that mitochondrial uncoupling appeared not to significantly affect calcium oscillations in either progesterone- or nitric oxide-stimulated human sperm [46]. Further investigations are needed to address the issue.

### 3.5. Mitophagy and Male Fertility

Mitophagy is an important cytoprotective “house-cleaning” mechanism that regulates the quality of mitochondria and maintains the perpetual renewal of mitochondria as well as cellular homeostasis [47]. Mitochondria are ubiquitinated during spermatogenesis, which not only promotes the degradation of defective sperm in the epididymis, but also that of sperm mitochondria after fertilization [48]. In fertilized mammalian eggs, the targeted clearance of sperm mitochondria may involve at least three pathways related to mitophagy and the ubiquitin–proteasome system (UPS).

Recent evidence has indicated that mitophagy plays an important role in maintaining spermatogenic cell homeostasis and maturation [49]. As the degree of asthenspermia worsens, mitochondrial morphology, distribution, and function change, and the number of normal mitochondria gradually decreases. Under these conditions, mitophagy may exert a beneficial effect on sperm motility [50]. In conclusion, the role of mitophagy is to promptly remove damaged mitochondria and protect spermatogenic cells from mitochondrial dysfunction, thereby reducing spermatogenic cell apoptosis. Decrease in mitophagy may accelerate the aging process.

### 3.6. Mitochondria-Mediated Apoptosis and Male Fertility

The role of mitochondria in cell apoptosis has been widely documented. When mitochondrial dysfunction occurs, the mitochondrial membrane permeability transition pore (mPTP) opening is prolonged, resulting in the release of cytochrome C, apoptosis-inducing factor (AIF), Ca^2+^, and other apoptosis-associated factors into the cytoplasm. This is followed by the activation of key members of the apoptosis-related caspase protein family, the destruction of nuclear chromatin, or effects on other Ca^2+^-dependent proteins, resulting in the disruption of the cellular structure and the promotion of cell apoptosis [51]. Numerous studies have shown that the markers of apoptosis in mammalian sperm are similar to those in somatic cells, such as the externalization of phosphatidylserine (PS) on the plasma membrane, the loss of mitochondrial integrity, activation of caspase, and DNA damage [6]. Although all these observations suggest that apoptosis indeed exists in sperm, the lack of cytoplasmic components in sperm and the distance between the nucleus and mitochondria render this possibility controversial.

Studies have confirmed that mitochondria-mediated sperm apoptosis can lead to ab-normal ATP metabolism and the disintegration of the sperm cell membrane, ultimately leading to decreased sperm count and infertility. Interestingly, markers of apoptosis have been detected in the sperm of infertile males and immature sperm, but not in the mature sperm of fertile males [6]. Meanwhile, ROS levels, cytochrome C release, and the activation of caspase-9 and -3 were reported to be positively correlated with male infertility [52] and altered sperm function [53]. Studies have found that sperm MMP is significantly reduced, while the apoptosis rate is significantly increased; consequently, sperm motility goes down. Thus, the increased sperm apoptosis appears to be the key factor leading to low sperm motility. In addition, signaling via the mitochondrial apoptosis pathway can damage DNA in the sperm nucleus, ultimately affecting the normal fertilization process and resulting in abnormal embryo implantation after fertilization [6]. Significantly negative correlations were observed between the age of the men and the percentage of alive spermatozoa. Thus, advanced male age is associated with increased sperm apoptosis [54,55].

### 3.7. Age-Related Male Infertility and the Changes of Sperm Mitochondria

With the increase in male fertility age worldwide, more and more studies have begun to focus on the impact of advanced paternal age (APA) on male fertility and offspring health. The most commonly used criterion to define advanced paternal age (APA) is age > 40 years at conception [11]. The aging of advanced male germ cells is multifaceted (Figure 3), including decreased semen volume, sperm count, motility, and normal morphology, increased sperm DFI and methylation, all of which may lead to male infertility and adverse effects on offspring [56,57].

A systematic meta-analysis examining the effects of aging on semen parameters found that men over 50 years experienced a 3–22% decrease in semen volume, 3–37% in sperm motility, and a 4–18% decrease in normal morphology compared with men under 30 years. Meanwhile, the pregnancy rate in the advanced age group was relatively reduced by 23–38% [58]. Another study in 2013 analyzed the semen data of 5081 men aged 16.5–72.3 years and found that the semen parameters did not change significantly before the age of 34, while the total sperm count, sperm motility, and normal morphology decreased year by year [59]. Studies also showed similar results that male aging was accompanied by a decrease in semen volume, total sperm count, sperm motility, normal-morphed sperm, and an increase in DFI [60]. A latest systematic review found that APA was associated with increased DFI, which suggested that DFI should be assessed in infertile older men with normal semen parameters to better identify the etiology of infertility [61]. Another study in 2013 analyzed sperm 5-methylcytosine (5-mC) and 5-hydroxymethylcytosine (5-hydroxymethylcytosine, 5-hmC) levels and found that sperm DNA methylation status was stable in the short term, but increased with age, with an average annual increase of 1.76% for 5-mC and nearly 5% for 5-hmC. The above studies together show that male fertility gradually declines with age [62].

Mitochondrial structure and function alter during aging. Aging can lead to a decrease in sperm MMP and cause oxidative damage to mtDNA [12,63]. Studies have also confirmed that the antioxidant capacity of aging male sperm was reduced, while the levels of ROS and lipid peroxides were increased [11,64]. In addition, mitochondria in hamsters undergo significant morphological change with aging, including mitochondrial swelling, mitochondrial vacuolization, and significantly reduced cristae [65].

## 4. The Application of Antioxidants in the Treatment of Male Infertility

Mitochondrial ROS signaling constitutes a significant part of the cellular processes regulated by organelles in sperm. Physiological ROS concentrations are essential for proper sperm function, whereas excessive ROS production overwhelming the antioxidant defense system can induce OS. This results in the decline or even the loss of sperm MMP, damaging the mitochondrial structure, and consequently reducing sperm motility which increases the occurrence of male infertility [66]. Mitochondria are involved in the biosynthesis of steroid hormones, while OS can reduce testosterone secretion through its effects on Leydig cells [17]. Moreover, ROS are the intermediates factors in sperm apoptosis. Excessive ROS can induce the abnormal opening of the mPTP and the release of apoptosis-related factors, thus accelerating sperm apoptosis and leading to male infertility (Figure 4). OS has also been implicated in aging processes and may represent one of the major causes of aging-related fertility loss.

Given that loss of male fertility is highly associated with OS, clinicians tend to administer different types of antioxidants to enhance the scavenging ability of the antioxidant defense system, thus improving sperm vitality and function. Antioxidants are ROS “scavengers” which help maintain the body’s oxidant/antioxidant balance (Figure 5) [67]. Antioxidants are mainly classified into two types—enzymatic and non-enzymatic. Enzymatic antioxidants mainly include superoxide dismutase (SOD), catalase (CAT), glutathione peroxidase (GPx), and glutathione reductase (GR). Glutathione (GSH), *N*-acetylcysteine (NAC), melatonin, coenzyme Q10 (CoQ10), vitamins C and E, folic acid, carnitine, zinc, selenium, and lycopene constitute the primary non-enzymatic antioxidants. Multiple studies have confirmed that the administration of exogenous antioxidants can counteract oxidative damage or OS, thereby improving sperm motility and DNA integrity in infertile men (Table 1).

### 4.1. Enzymatic Antioxidants

SOD is one of the most effective antioxidant enzymes in the body and is also the first line of defense against excess ROS production. This enzyme can transform superoxide anion free radicals into H_2_O_2_, and then convert H_2_O_2_ into H_2_O and O_2_ through GPx or CAT. Studies have shown that SOD can protect sperm from lipid peroxidation and OS [21].

CAT, which is derived from the prostate, has been detected in human and rat seminal plasma and sperm cells, and plays a key role in nitric oxide-induced sperm capacitation [17]. Another enzyme acting in the semen antioxidant system is GPx, which is also de-rived from the prostate. GPx can reduce the ROS content by catalyzing the reduction in hydrogen peroxide and organic peroxides (including phospholipid peroxides). In sperm, GPx is mainly located in the mitochondrial matrix, but a nuclear form that protects sperm DNA from oxidative damage and participates in chromatin enrichment has also been found [23].

### 4.2. Non-Enzyme Antioxidants

Low levels of GSH in human seminal plasma can lead to abnormal sperm flagellar structure and eventually cause dyskinesia [17]. GSH supplementation can significantly improve the sperm parameters (such as sperm number, motility, and morphology) of infertile men [68,69,70]. NAC, the precursor of GSH, can improve semen volume, viscosity, and motility, as well [71,72]. Animal experiments have confirmed that NAC can protect sperm DNA from oxidative damage and improve sperm function [73].

Melatonin, a neuroendocrine hormone secreted by the pineal gland, has strong antioxidant capacity, can induce the production of antioxidant enzymes, and directly remove various oxygen free radicals. Melatonin can regulate testicular development and reduce testicular damage, and can thus be used for the preservation of male fertility in the clinic. In recent years, melatonin has also been widely used as a cryoprotectant for sperm. Melatonin plays important roles in maintaining the fluidity and vitality of sperm membranes [74]. One study reported that the DFI of sperm was significantly reduced, and the quality of embryos was improved in infertile men who took 6 mg of melatonin daily and received IVF treatment 90 days later [75]. Another study of infertile men after varicocele resection found significant improvement in routine semen parameters (sperm concentration, motility, and morphology) and total antioxidant capacity (TAC) after the oral administration of 400 mg of melatonin daily for 3 months [76].

CoQ10 is abundantly present in the middle piece of sperm flagella and can promote sperm maturation and improve sperm quality by promoting mitochondrial electron transfer. CoQ10 is a fat-soluble antioxidant, and studies have found that CoQ10 can also regenerate other antioxidants such as vitamin C and E, thereby augmenting its antioxidant capacity [77]. A prospective study suggested that the treatment of male infertility with CoQ10 (200 mg/day for 6 months) elicited a significant increase in sperm concentration and motility [78]. A different study found that oral CoQ10 treatment for idiopathic oligoasthenozoospermia (200 mg/day for 3 months) significantly increased the TAC in seminal plasma, as well as sperm concentration and motility [79].

Vitamin C (ascorbic acid, water-soluble) and vitamin E (α-tocopherol, fat-soluble) are the most common vitamin antioxidants, and both can neutralize and reduce ROS, thereby attenuating OS-induced damage [21]. The content of vitamin C in human seminal plasma is approximately 10-fold that in serum. The concentration of vitamin C is reported to be positively correlated with the percentage of normal sperm and negatively correlated with DFI [80]. Clinical studies have found that oral vitamin C administered at the dosage of 1 g/day can effectively treat male infertility [81]. Vitamin E can protect sperm cell membranes from OS and prevent lipid peroxidation [82]. A clinical trial showed that a daily intake of 600 mg of vitamin E improved sperm function after 3 months [17].

Folic acid (vitamin B9, water-soluble) is involved in nucleic acid synthesis and amino acid metabolism. Its application in the treatment of male infertility is based on its ability to scavenge free radicals. A meta-analysis in 2017 evaluated the effects of the combined use of folic acid and zinc on endocrine parameters and sperm in infertile men, and found that this drug combination exerted a beneficial effect on sperm quality in men with low fertility [83]. However, another large-scale randomized clinical trial in 2020 reported that the administration of a folic acid (5 mg/day) plus zinc (30 mg/day) supplement to infertile men for 6 months did not significantly improve semen quality or the live birth rate of assisted reproductive technology (ART)-induced pregnancies [84].

Carnitine is also known as a “quasi-vitamin”. Both L-carnitine (LC) and L-acetyl carnitine (LAC) are water-soluble antioxidants that can participate in sperm metabolism and promote sperm motility [2,20]. One study found that LC can improve human sperm motility and vitality, but has no effect on the oxidation of sperm DNA during cryopreservation [85]. Studies have also confirmed that carnitine content is significantly reduced in the semen samples of infertile men, and carnitine intervention can reduce ROS levels and improve semen quality in infertile men [86]. The latest meta-analysis found that the LC and LAC combination represents an effective treatment for men with idiopathic oligoasthenoteratozoospermia [87].

Zinc is an important and the second most abundant microelement in human. It is versatile and involved in a broad array of molecular events in the male reproductive system. Zn, acting as a heavy metal detox, an antibacterial agent, and a hormone balancer, plays key roles in regulating sexual health and functions [88]. Zn is also an important antioxidant and can improve multiple sperm parameters [89]. The concentration of zinc in seminal fluid is approximately 30 times higher than in blood. This helps to maintain the semen quality and homeostasis of testicular environment [90]. Other studies also confirmed that the zinc concentration in seminal plasma of fertile men is significantly higher than that of infertile men [91]. Zinc’s antioxidant properties appear to be attributed to its ability to reduce production of hydrogen peroxide and hydroxyl radicals by antagonizing redox-active transition metals (such as iron and copper) [92]. Given the versatile and critical roles, it is very essential to understand its multifunctionality and the mechanisms underlying the complicated biological processes in the future.

**Table 1 pharmaceuticals-15-00519-t001:** Application of non-enzymatic antioxidants in the treatment of male infertility.

Type	Function	Administration	Common Dosages	References
Glutathione	Enhances enzymatic antioxidant activity	Intramuscular Oral	600 mg every other day 100 mg/day	[68,69] [70]
*N*-acetylcysteine	Enhances enzymatic antioxidant activity and scavenges free radical	Oral	600 mg/day	[71,72]
Melatonin	Activates the production of antioxidant enzyme and scavenges free radicals	Oral	6 mg/day 400 mg/day	[75] [76]
Coenzyme Q10	Scavenges free radicals in a reduced form in the mitochondrial electron transport system	Oral	200 mg/day	[78,79]
Vitamin C	Neutralizes free radicals	Oral	500–1000 mg/day	[67,81]
Vitamin E	Neutralizes free radicals	Oral	200–1000 mg/day	[17,67]
Folic acid (vitamin B9)	Scavenges free radicals	Oral	5 mg/day	[83,84]
Carnitines	Neutralize free radicals and serve as an energy source	Oral	500–1000 mg/day	[20,67]
Zinc	Inhibits NADPH oxidase	Oral	30–500 mg/day	[83,84,89]
Selenium	Enhances enzymatic antioxidant activity	Oral	50–200 μg/day	[2,67]
Lycopene	Neutralizes free radicals	Oral	6–25 mg/day	[2,67]

Selenium is another important microelement involved in spermatogenesis. Selenium protects sperm DNA from OS-induced damage and helps maintain sperm structural integrity [93]. Lycopene is a natural carotenoid widely found in fruits and vegetables and has strong ROS quenching ability. Lycopene concentrations are high in human testes and seminal plasma but appear to be lower in infertile men [94].

A systematic meta-analysis carried out in 2018 included a total of 19 retrospective randomized controlled studies and 10 prospective randomized cohort studies, 26 of which reported that antioxidant therapy improved routine semen parameters and ART outcomes. However, the question of whether antioxidants can treat male infertility remains controversial. It was reported that oral antioxidants cannot improve semen quality and are not conducive to improving the pregnancy rate of the partners of male infertile patients [95]. On the other hand, a Cochrane study in March 2019, which covered 17 antioxidants, reported that antioxidants could increase live birth rates in the offspring of infertile male patients [20]. Further in-depth research is needed to resolve this disagreement.

### 4.3. Antioxidants in the Treatment of Aging Infertile Men

There are many factors that cause the decline of semen quality in aging males, one of which is the toxic effect of OS [96,97]. Mitochondria and sperm plasma membrane are the two major sites for ROS generation in sperm. With the increase in age, the human antioxidant capacity gradually declines, and the generated ROS cannot be effectively removed to induce OS, resulting in lipid peroxidation, DNA fragmentation, enzymatic denaturation, and damage to sperm mitochondrial structure and function. The damaged mitochondria will produce more ROS and fall into a vicious circle, resulting in the decline of fertility in aging males [98,99]. Then, antioxidant therapy for aging males should be effective.

Indeed, there are only a small number of studies on antioxidant treatments specifically for aging males (Table 2). Some studies have found that antioxidants can improve sperm quality in aging males [100,101] and some have found no improvement [100]. In addition, in vitro studies have found that adding idebenone (a mitochondrial-permeable synthetic benzoquinone that acts as an antioxidant by scavenging free electrons) to the sperm of men over 40 years old can reduce sperm ROS and improve the fertilization rate [102]. However, the sample size of these studies is small, and the combination of multiple antioxidants is generally used, which cannot effectively prove the specific mechanism of antioxidants.

The overall effectiveness of using antioxidants to treat male infertility is currently uncertain. Aggressive antioxidant supplementation may also cause side effects, especially during ART. Additionally, experts have emphasized that the oxidation status of infertile male patients should be evaluated before the use of antioxidants is recommended. Therefore, more double-blind, randomized, large-sample, and multicenter trials are needed to determine the dose and duration of the clinical use of antioxidants.

Some antioxidants did not show any improvement of sperm quality after oral administration. This could be due to the inefficient absorbance of antioxidants in tissue. Most antioxidants can cross the plasma membrane and scavenge the toxic consequences of ROS in the cells, however, it is important to test whether the dietary and orally administered antioxidants become absorbed by the target tissue to the therapeutic levels and to evaluate the effects. With the wide application of antioxidants, the determination methods of antioxidant effects and abilities are constantly developing and improved. Given that male infertile patients usually show higher levels of ROS and malondialdehyde (MDA), accompanied by a decrease in TAC, the clinical evaluation of antioxidant effects is generally performed by chemiluminescence or colorimetry to detect seminal plasma ROS, MDA, and TAC concentrations [2,75,76]. However, these methods all measure the anti-radical capacity of antioxidants in the final state of the reaction. Oxygen radical absorbance capacity (ORAC) is an exciting and revolutionary analytical method based on fluorescence decay, which can dynamically monitor the process of antioxidants inhibiting free radical chain reactions [103,104]. The advantages of this method are close to the body physiological conditions, complete chemical reaction, simple operation and high sensitivity, and not easily disturbed by human factors. The study in 2020 evaluated the redox status of non-obstructive azoospermia (NOA) patients through ORAC, and found that the production of ROS may be directly related to spermatogenesis disorders [105]. Unfortunately, no relevant studies of ORAC application to detect the effect of antioxidants in treating male infertility were retrieved.

## 5. Conclusions

Recently, the role of mitochondria in reproduction, especially their importance for gamete quality, has opened up a new frontier in both fundamental research and clinical studies. Although paternal mitochondria are mostly eliminated after fertilization and may seem unimportant, they are essential for normal sperm physiological function and fertilization. Mitochondria in sperm have unique structures and physiological functions. Not only producing ATP, mitochondria are also involved in ROS production, calcium signal transduction, mitophagy, and apoptosis, which are necessary for sperm motility, capacitation, and acrosome response, as well as fertilization. These processes are inter-related. It may be needed to uncouple them and characterize the roles each one plays in male reproduction. Such further investigations will also help tease out the missing links among these molecular events and offer new insights into the mechanisms of mitochondrial regulation on male fertility.

OS due to excessive ROS production is a key factor in the destruction of sperm mitochondria. OS cannot only directly destroy mitochondrial structure, but also damage the ETC complex and mitochondrial DNA (mtDNA), leading to a decrease in MMP, prolonged mPTP opening, and the occurrence of cell apoptosis. In turn, these phenomena will promote ROS production, eventually leading to a vicious cycle which results in impaired sperm quality and male infertility. Mitochondrial oxidation is the major source of oxidative lesions that accumulate with age. Increasing evidence has been suggesting that mitochondria are the common link between age and age-related loss of male fertility. A healthy lifestyle and physical exercise are crucial for reducing OS and avoiding male infertility. Given that the balance between pro- and antioxidative agents is often shifted towards the pro-oxidizing condition in aging testis mitochondria, and the OS is a key factor impairing sperm function, antioxidant interventions hold great potential as promising therapeutic strategies to attenuate the negative effects of aging (and the resulting oxidative stress) on the male reproductive system. However, many current studies are performed in rodent animal models or are based on correlational research. More extensive studies and analyses may be required to evaluate the long-term effects in various settings. Particularly, owing to the difference of reproductive mechanisms between rodents and humans, and the possible issues caused by interspecies dosage scaling, investigations using primate animal models may help better evaluate the therapeutic effects. In addition, the optimal antioxidant regimen that can offer safe and efficient therapy in clinical practice needs to be identified in further studies. There is a long way to go, but further investigations will contribute to the development of effective therapeutic approaches to treat male infertility.

## Figures and Tables

**Figure 1 pharmaceuticals-15-00519-f001:**
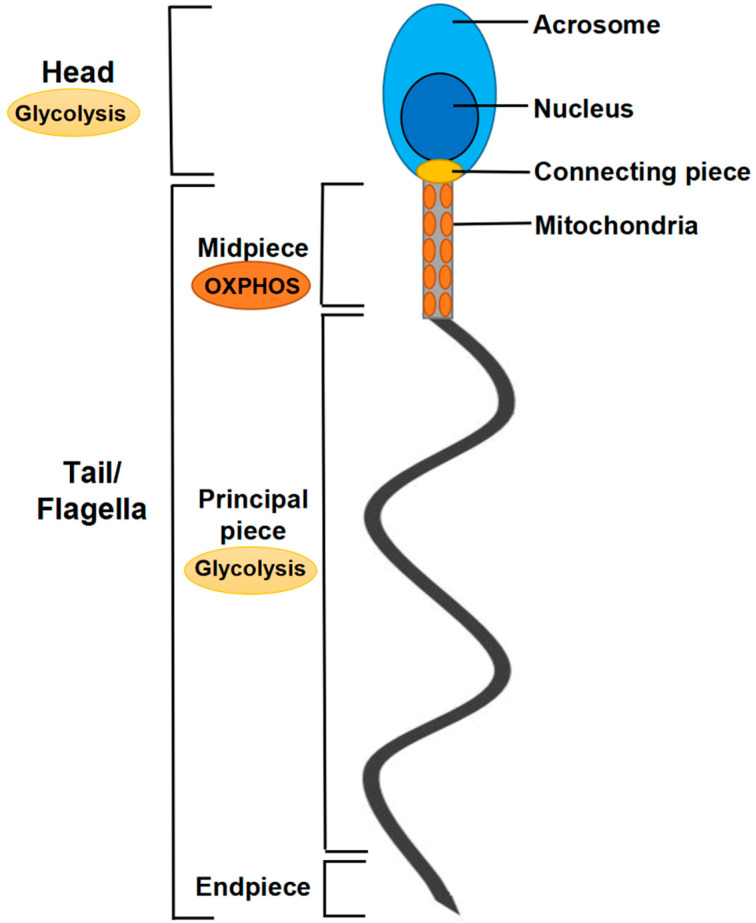
Schematic diagram of human sperm structure. The sperm head is composed of acrosome and nucleus, which is also the site of glycolysis. The tail/flagellum region contains connecting piece, midpiece, principal piece, and endpiece. Mitochondria are located in midpiece, which is the site of OXPHOS.

**Figure 2 pharmaceuticals-15-00519-f002:**
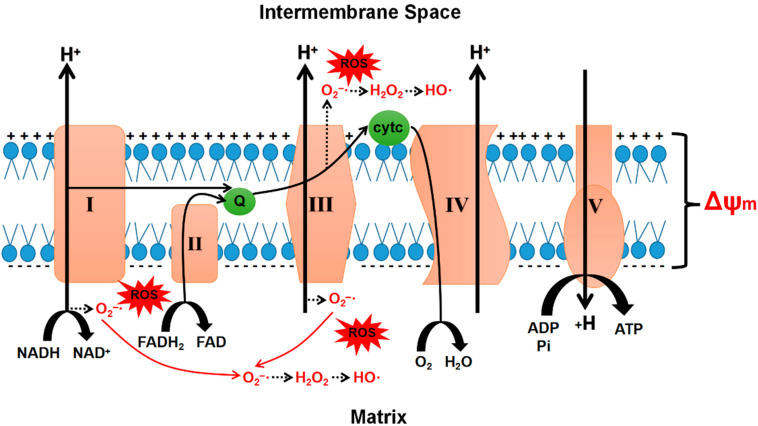
Schematic diagram of mitochondrial ETC and ROS production. ETC consists of four multimeric complexes (I–IV). Complexes I, III, and IV pump protons from the matrix into the intermembrane space, forming a transmembrane potential difference (Δψm). Complexes I and III are the main ROS-generating subunits, and superoxide is released into the matrix from these two complexes as well as the intermembrane space of complex III. Oxygen is reduced to water at complex IV, which is the main oxygen consuming step of cellular respiration. Complex V, ATP synthase, is the place where ATP is synthesized by utilizing the proton-motive force created across the inner mitochondrial membrane.

**Figure 3 pharmaceuticals-15-00519-f003:**
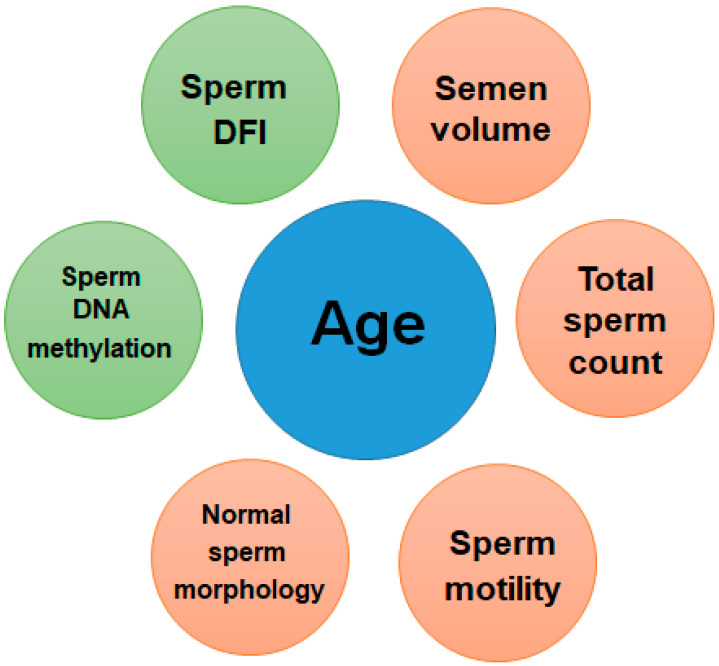
Age-related changes in sperm quality. Different colors denote the type of association with age: positive association in green and negative association in orange.

**Figure 4 pharmaceuticals-15-00519-f004:**
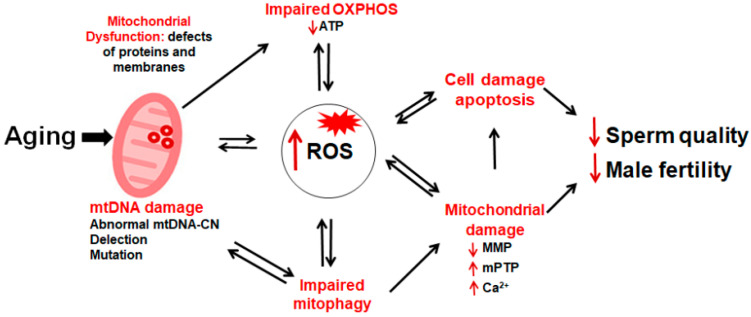
Schematic diagram of mitochondrial dysfunction and male infertility. Aging of the male reproductive system involves multi-factorial changes. Alterations to mitochondria are among the most remarkable features contributing to lower fertility. Abnormal mitochondrial OXPHOS leads to a decrease in ATP production, and abnormal mitophagy results in the inability to clear damaged mitochondria in time, all of which will lead to an increase in ROS. Excessive ROS in turn will further affect mitochondrial function, and eventually fall into a “vicious circle”, leading to mitochondrial damage and cell apoptosis, affecting sperm quality and fertilization potential, and ultimately contributing to male infertility.

**Figure 5 pharmaceuticals-15-00519-f005:**
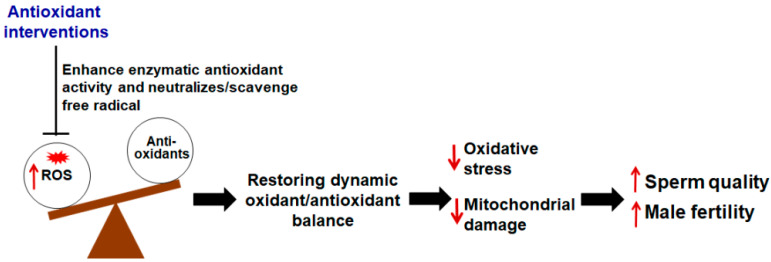
Value of antioxidants in the treatment of male infertility. Antioxidants are ROS “scavengers” which help maintain the body’s oxidant/antioxidant balance. Antioxidative interventions can counteract oxidative damage or OS and reduce mitochondrial damage, thereby improving sperm quality and male fertility.

**Table 2 pharmaceuticals-15-00519-t002:** Application of antioxidants in the treatment of aging men.

Antioxidant Treatment	Sample	Findings	References
Zinc and folate	Aged men (>40 years), 57 cases	No improvement in semen quality	[100]
Vitamins C, E and beta-carotene	Aged men (>40 years), 57 cases	Improvement in sperm numbers and motility	[100]
Vitamin C and E, zinc	Aged men (>44 years), 34 cases	20% less sperm DNA damage.	[101]
Idebenone	Aged men (>40 years), 7 cases	In vitro addition of 5 µM and 50 µM idebenone reduced sperm ROS concentration and increased fertilization rates	[102]

## Data Availability

Not applicable.

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
