# Peer review of "Age-Related Decline of Male Fertility: Mitochondrial Dysfunction and the Antioxidant Interventions"

_pharmaceuticals, 2022, doi:10.3390/ph15050519_

Round 1
Reviewer 1 Report
The review article entitled “Mitochondrial dysfunction, age-related decline of male fertility, and the antioxidant interventions” deals with the physiological importance of sperm mitochondria in fertility.
The manuscript is well structured; it contains well-done figures that summarize sections of the article. It covers the topic very well, although it does not go very deep, but it provides basic concepts in a correct way.
A section that comments on the relevance of seminal fluid in sperm function would improve the manuscript, for example: the hormones it contains, antioxidant activity, energetic compounds present, etc.
The authors must review reference 44, which does not seem to correspond to what is commented in the manuscript.
Author Response
We thank the reviewers for their insightful suggestions and comments on the manuscript.
Following the suggestions from the reviewers, we, accordingly, have made multiple revisions in the figures and the text throughout. The major changes were highlighted in red. Altogether, these revisions have substantially improved the manuscript, which we hope is now suitable for publication in Pharmaceuticals.

Reviewer 2 Report
Major Revision:
1. The key points should be focused nicely in the 'Introduction' section. Need to change the introduction section considerably. Try to include the existing research limitations and state how the present research unravels those limits.
2. The current title is confusing. Please change into more straight form.
3. Normal sperm structure and function underlie normal male fertility. Need references.
4. Introduction should be improved. Authors need to add references during mentioning statements.
5. Mitochondria in Spermatozoa: Authors add more recent insights with appropriate references.
6. Structural Changes in Mitochondria need to mention.
7. Clinical Aspects can add to increase the value.
8. Number of references is not enough as a review article.
9. Why do authors only focus on male fertility?
10. The conclusion needs to address future perspectives.
11. 4.2. Non-Enzyme Antioxidants: Authors need to add more insights by adding the table.
12. More mechanistic figures should need to add (at least two).
13. Role of antioxidants and zinc is unclear in treating male infertility—more precise information needs to be added. Need more specific information’s how mitochondrial structure is changed with age-related decline.
14. Extensive language issue which needs expert attention.
15. Authors must check the references and rearrange them according to the Journal Guidelines.
Author Response
We thank the reviewers for their insightful suggestions and comments on the manuscript.
Major Revision:
- The key points should be focused nicely in the 'Introduction' section. Need to change the introduction section considerably. Try to include the existing research limitations and state how the present research unravels those limits.
Response: Thanks for the suggestion. We have revised the induction. The section in the revised manuscript includes three major parts: 1.) sperm structure and the distribution of mitochondria in sperm, 2.) the general background of altered mitochondrial function in male fertility declines during aging 3.)seminal fluid and the antioxidants in the seminal plasma& beneficial effects of antioxidants.
Although “Several clinical studies have reported that oral antioxidants can help reduce OS and improve male fertility”, many of these results are not conclusive. So, we mentioned that “it is still necessary to evaluate the long-term effects in various settings”. In our “Conclusion” section, we also suggested that “owing to the difference of reproductive mechanisms between rodents and humans, and the possible issues caused by interspecies dosage scaling, investigations using primate animal models may help better to evaluate the therapeutic effects. In addition, optimal antioxidant regimen that can offer safely and efficient therapy in clinical practice needs to be identified in the further studies” .
- The current title is confusing. Please change into more straight form.
Response: Thanks for the suggestion. We have revised the title. ( “Age-related decline of male fertility: mitochondrial dysfunction and the antioxidant interventions”)
- Normal sperm structure and function underlie normal male fertility. Need references.
Response: Thanks for the suggestion. References have been added as suggested.
- Introduction should be improved. Authors need to add references during mentioning statements.
Response: Thanks for the suggestion. We have revised the introduction and added the references as suggested. The introduction section in the revised manuscript now includes three major parts: 1.) sperm structure and the distribution of mitochondria in sperm, 2.) the general background of altered mitochondrial function in male fertility declines during aging 3.)seminal fluid and the antioxidants in the seminal plasma & beneficial effects of antioxidants .
- Mitochondria in Spermatozoa: Authors add more recent insights with appropriate references.
Response: Thanks for the suggestion. We have revised the part as suggested. More recent references are included.
“…Thus, mitochondrial fusion and fission (mitochondrial dynamics) realizes the dynamic changes in mitochondrial number and structure during spermatogenesis, allowing fine-tuned responses to cellular energy demands[6,25].”
“In addition, through mitochondrial dynamics, mitochondria in mature sperm are highly concentrated, and the mitochondrial metabolism is more efficient at this time, which also plays a key role in maintaining sperm motility and fertilization[22,26]. However, the role of mitochondrial dynamics during spermatogenesis and sperm maturation remains largely unknown.”
These recent references are included.
Boguenet, M.; Bouet, P.E.; Spiers, A.; Reynier, P.; May-Panloup, P. Mitochondria: their role in spermatozoa and in male infertility. Hum Reprod Update 2021, 27, 697-719, doi:10.1093/humupd/dmab001.
Varuzhanyan, G.; Chan, D.C. Mitochondrial dynamics during spermatogenesis. J Cell Sci 2020, 133, doi:10.1242/jcs.235937.
Park, Y.J.; Pang, M.G. Mitochondrial Functionality in Male Fertility: From Spermatogenesis to Fertilization. Antioxidants (Basel) 2021, 10, doi:10.3390/antiox10010098.
Gu, N.H.; Zhao, W.L.; Wang, G.S.; Sun, F. Comparative analysis of mammalian sperm ultrastructure reveals relationships between sperm morphology, mitochondrial functions and motility. Reprod Biol Endocrinol 2019, 17, 66, doi:10.1186/s12958-019-0510-y.
- Structural Changes in Mitochondria need to mention.
Response: Thanks for the suggestion. We have included the part as suggested in the revised manuscript.
“Mitochondrial structure and function alter during aging. Aging can lead to a decrease in sperm MMP and cause oxidative damage to mtDNA[12,63]. Studies have also confirmed that the antioxidant capacity of aging male sperm was reduced, while the level of ROS and lipid peroxides were increased[11,64]. In addition, mitochondria in hamsters undergo significant morphological change with aging, including mitochondrial swelling, mitochondrial vacuolization, and significantly reduced cristae[65].”
- Clinical Aspects can add to increase the value.
Response: Thanks for the suggestion. These are really good suggestions. We have included the part in the revised manuscript.
We included Clinical results and aspects in the sections of “Age-related male infertility and the changes of sperm mitochondria” and Conclusion.
“With the increase of male fertility age worldwide, more and more studies begin to focus on the impact of advanced paternal age (APA) on male fertility and offspring health. The most commonly used criterion to define advanced paternal age (APA) is age > 40 years at conception[11]. The aging of advanced male germ cells is multifaceted (Figure 3), including decreased semen volume, sperm count, motility and normal morphology, increased sperm DFI and methylation, all of which may lead to male infertility and adverse effects on offspring[56,57]. ”
“Another study in 2013 analyzed the semen data of 5081 men aged 16.5-72.3 years and found that the semen parameters did not change significantly before the age of 34, while the total sperm count, sperm motility, and normal morphology decreased year by year[59]. Studies also showed similar results that male aging was accompanied by a decrease in semen volume, total sperm count, sperm motility, normal-morphed sperm, and an increase in DFI[60].”
“Another study in 2013 analyzed sperm 5-methylcytosine (5-mC) and 5-hydroxymethylcytosine (5-hydroxymethylcytosine, 5-hmC) levels and found that sperm DNA methylation status was stable in the short term, but increased with age, with an average annual increase of 1.76% for 5-mC and nearly 5% for 5-hmC. The above studies, together,show that male fertility declines gradually with age[62].
Mitochondrial structure and function alter during aging. Aging can lead to a decrease in sperm MMP and cause oxidative damage to mtDNA[12,63]. Studies have also confirmed that the antioxidant capacity of aging male sperm was reduced, while the level of ROS and lipid peroxides were increased[11,64]. In addition, mitochondria in hamsters undergo significant morphological change with aging, including mitochondrial swelling, mitochondrial vacuolization, and significantly reduced cristae[65]. ”
“Given that the balance between pro- and anti-oxidative agents is often shifted towards the prooxidizing condition in aging testis mitochondria, and the OS is a key factor impairing sperm function, antioxidant interventions hold great potential as the promising therapeutic strategies to attenuate the negative effects of aging (and resultant oxidative stress) on the male reproductive system. However, many of current studies are performed in rodent animal models or are based on correlational research. More extensive studies and analyses may be required to evaluate the long-term effects in various settings. Particularly, owing to the difference of reproductive mechanisms between rodents and humans, and the possible issues caused by interspecies dosage scaling, investigations using primate animal models may help better to evaluate the therapeutic effects. In addition, optimal antioxidant regimen that can offer safely and efficient therapy in clinical practice needs to be identified in the further studies.”
- Number of references is not enough as a review article.
Response: Thanks for the suggestion, the references have been supplemented.
- Why do authors only focus on male fertility?
Response: It is an interesting topic. The roles of mitochondria in male reproduction have remained an enigma. Mitochondrial alterations are associated with male fertility decline and have been implicated in ageing. Lots of studies support that pathophysiologic relevance, yet its existence as a causative factor needs to be further characterized. We here aim to review the major existing literatures, and summarize the progress made in understanding the mitochondrial dysfunction, aging and male infertility, and expect to provoke new insights into the pathophysiologic relevance and underlying mechanisms.
It is also an important topic. Approximately 50% of cases of infertility are partially or wholly attributable to male factors. The clinical potential of antioxidant interventions is also discussed in the present review. Hopefully, such discussion on this topic can provide new ideas for the treatment of the disroders.
- The conclusion needs to address future perspectives.
Response: Thanks for the suggestion, We included perspectives from both biologic and clinic aspects in the Conclusion part.
“These processes are inter-related. It may be needed to uncouple them and characterize the roles each one plays in male reproduction. Such further investigations will also help to tease out the missing links among these molecular events and offer new insights into the mechanisms of mitochondrial regulation on male fertility.”
“…many of current studies are performed in rodent animal models or are based on correlational research. More extensive studies and analyses may be required to evaluate the long-term effects in various settings.”
“Particularly, owing to the difference of reproductive mechanisms between rodents and humans, and the possible issues caused by interspecies dosage scaling, investigations using primate animal models may help better to evaluate the therapeutic effects. In addition, optimal antioxidant regimen that can offer safely and efficient therapy in clinical practice needs to be identified in the further studies.”
- 4.2. Non-Enzyme Antioxidants: Authors need to add more insights by adding the table.
Response: Thanks for the suggestion. We have added the table.
- More mechanistic figures should need to add (at least two).
Response: Thanks for the suggestion. We have added 2 Figures.
- Role of antioxidants and zinc is unclear in treating male infertility—more precise information needs to be added. Need more specific information’s how mitochondrial structure is changed with age-related decline.
Response: Thanks for the suggestion. We included more description in the revised manuscript as suggested.
“Zinc is an important and the second most abundant microelement in human.It is versatile and is involved in a broad array of molecular events in the male reproductive system. Zn, acting as a heavy metal detox, an antibacterial agent and a hormone balancer, plays key roles in regulating sexual health and functions[88]. Zn is also an important antioxidant and can improve multiple sperm parameters[89].”
“…The concentration of zinc in seminal fluid is approximately 30 times higher than in blood. This helps to maintain the semen quality and homeostasis of testicular environment[90]…”
“Zinc’s antioxidant properties appears to be attributed to its ability to reduce production of hydrogen peroxide and hydroxyl radicals by antagonizing redox-active transition metals (such as iron and copper)[92]. Given the versatile and critical roles, it is very essential to understand its multifunctionality and the mechanisms underlying the complicated biological processes in the future. ”
- Extensive language issue which needs expert attention.
Response: Thanks for the suggestion. The manuscript has been reviewed and edited by an English language expert.
- Authors must check the references and rearrange them according to the Journal Guidelines.
Response: Thanks for the suggestion. We have rearranged the references according to the Journal Guidelines.
Reviewer 3 Report
In this review, the authors examined the role of mitochondria and ROS-triggered signaling in the age-related male fertility. Although there don’t provide musch novel viewpoints to discuss, it is still worth mentioning that the author has compiled relevant reports about antioxidants therapy in male fertility diseases. Overall the manuscript is a well-written, and compact piece of work. The topic chosen by the authors is worth writing a review article. The titles, organization of contents, English language are acceptable. References cited in the manuscript are latest and up to date.
Major to be revised:
The references from 44-52 cited in Table 1 are all mistyping. The authors need further correct it again.
Author Response
We thank the reviewers for their insightful suggestions and comments on the manuscript.
In this review, the authors examined the role of mitochondria and ROS-triggered signaling in the age-related male fertility. Although there don’t provide musch novel viewpoints to discuss, it is still worth mentioning that the author has compiled relevant reports about antioxidants therapy in male fertility diseases. Overall the manuscript is a well-written, and compact piece of work. The topic chosen by the authors is worth writing a review article. The titles, organization of contents, English language are acceptable. References cited in the manuscript are latest and up to date.
We thank the reviewer for the favorable comments.
Major to be revised:
The references from 44-52 cited in Table 1 are all mistyping. The authors need further correct it again
Response: Thanks for the suggestion, the references in table1 have been revised and improved.
Reviewer 4 Report
After reviewing the manuscript, I did not get a very positive impression. First of all, this is due to the fact that this topic is simply "worn out" and there are many hundreds of reviews on the topic of antioxidants and male infertility, and in general they are very similar, not having novelty, but simply copying each other, but in different journals (for example, this year there were similar, but deeper than this reviews, e.g., DOI: 10.3390/antiox11020306, DOI: 10.3389/fmolb.2021.799294, DOI: 10.3390/antiox11010167, DOI: 10.1089/ars.2021.0235, DOI: 10.1089/ars.2021.0238)
Therefore, I was looking for some novelty in comparison with this array of already published reviews, and I would be very grateful to the authors for pointing out what is the fundamental difference and what is the novelty in their consideration of the problem. I want to note that about 50% of cases of infertility are attributed to the presence of oxidative stress, so it would be nice to indicate all possible causes that are not caused by oxidative stress. This is my major point.
As minor comments, I draw attention to the frequent incorrect narrative and evaluation. For example:
Page 1 line 42. The cited work (3) does not cover all the aspects cited by the authors. It is better to use doi: 10.1023/a:1027304122259
Page 2 line 52. OXPHOS does not produce ROS. ROS is produced in the respiratory chain (ETC)
Page 3 line 90. What other organelles and cytoplasm are you talking about?
Page 3 line 100. What isoenzymes are you talking about
Page 3 line 111. Again, references (3, 17) do not correspond to the context and it is better to cite doi: 10.1023/a:1027304122259
Page 3 line 113. The word “mostly” is wrong and should be replaced by “exclusively”.
Page 5 line 188. The phrase is incorrect, because oxygen consumption is a complex function of the membrane potential. High oxygen consumption can be caused by depolarization of the mitochondrial membrane, therefore the word “positively” is true for the membrane potential and incorrect for oxygen consumption.
Page 6. Lines 202-203. The phrase is not accurate, because a certain increase in free Ca in the matrix will activate three dehydrogenases, however, at higher values, nonspecific permeability may occur, which is fatal.
Page 7. Line 243. The phrase is not correct. If apoptosis has already occurred, then the cell is dead and there can be no question of any motility.
Author Response
We thank the reviewers for their insightful suggestions and comments on the manuscript.
After reviewing the manuscript, I did not get a very positive impression. First of all, this is due to the fact that this topic is simply "worn out" and there are many hundreds of reviews on the topic of antioxidants and male infertility, and in general they are very similar, not having novelty, but simply copying each other, but in different journals (for example, this year there were similar, but deeper than this reviews, e.g., DOI: 10.3390/antiox11020306, DOI: 10.3389/fmolb.2021.799294, DOI: 10.3390/antiox11010167, DOI: 10.1089/ars.2021.0235, DOI: 10.1089/ars.2021.0238)
Therefore, I was looking for some novelty in comparison with this array of already published reviews, and I would be very grateful to the authors for pointing out what is the fundamental difference and what is the novelty in their consideration of the problem.
Response: Thanks for the suggestion. We have revised the manuscript as suggested. Comparing with the previously published reviews, it covers a broader scope. We included the several subtopics 1.) mitochondrial dysfunction and male infertility; 2.) altered mitochondrial function in male fertility declines during aging 3.) beneficial effects of antioxidants .
We added a section about” Age-related male infertility and the changes of sperm mitochondrial”:
“With the increase of male fertility age worldwide, more and more studies begin to focus on the impact of advanced paternal age (APA) on male fertility and offspring health. The most commonly used criterion to define advanced paternal age (APA) is age > 40 years at conception[11]. The aging of advanced male germ cells is multifaceted (Figure 3), including decreased semen volume, sperm count, motility and normal morphology, increased sperm DFI and methylation, all of which may lead to male infertility and adverse effects on offspring[56,57].
A systematic meta-analysis examining the effects of aging on semen parameters found that men over 50 years experienced a 3-22% decrease in semen volume, 3-37% in sperm motility, and a 4%-18% decrease in normal morphology compared with men under 30-years. Meanwhile, the pregnancy rate in the advanced age group was relatively reduced by 23-38%[58]. Another study in 2013 analyzed the semen data of 5081 men aged 16.5-72.3 years and found that the semen parameters did not change significantly before the age of 34, while the total sperm count, sperm motility, and normal morphology decreased year by year[59]. Studies also showed similar results that male aging was accompanied by a decrease in semen volume, total sperm count, sperm motility, normal-morphed sperm, and an increase in DFI[60]. A latest systematic review found that APA was associated with increased DFI, which suggested that DFI should be assessed in infertile older men with normal semen parameters to better identify the etiology of infertility[61]. Another study in 2013 analyzed sperm 5-methylcytosine (5-mC) and 5-hydroxymethylcytosine (5-hydroxymethylcytosine, 5-hmC) levels and found that sperm DNA methylation status was stable in the short term, but increased with age, with an average annual increase of 1.76% for 5-mC and nearly 5% for 5-hmC. The above studies, together,show that male fertility declines gradually with age[62].
Mitochondrial structure and function alter during aging. Aging can lead to a decrease in sperm MMP and cause oxidative damage to mtDNA[12,63]. Studies have also confirmed that the antioxidant capacity of aging male sperm was reduced, while the level of ROS and lipid peroxides were increased[11,64]. In addition, mitochondria in hamsters undergo significant morphological change with aging, including mitochondrial swelling, mitochondrial vacuolization, and significantly reduced cristae[65]. “
We also added more in-depth discussion and perspectives in the revised version.
“These processes are inter-related. It may be needed to uncouple them and characterize the roles each one plays in male reproduction. Such further investigations will also help to tease out the missing links among these molecular events and offer new insights into the mechanisms of mitochondrial regulation on male fertility. ”
“…However, many of current studies are performed in rodent animal models or are based on correlational research. More extensive studies and analyses may be required to evaluate the long-term effects in various settings.”
“Particularly, owing to the difference of reproductive mechanisms between rodents and humans, and the possible issues caused by interspecies dosage scaling, investigations using primate animal models may help better to evaluate the therapeutic effects. In addition, optimal antioxidant regimen that can offer safely and efficient therapy in clinical practice needs to be identified in the further studies.”
I want to note that about 50% of cases of infertility are attributed to the presence of oxidative stress, so it would be nice to indicate all possible causes that are not caused by oxidative stress. This is my major point.
Response: Thanks for the suggestion. We have revised the manuscript as suggested.
“There are many causes of male infertility, including genetic, endocrine, lifestyle and/or age-related factors. Some infections can interfere with sperm health or even block the passage of sperm. Varicoceles, ejaculation issues and defects of the tubules transporting sperm cause male infertility, too. Notably, many of the cases of male infertility involve OS. Approximately 50% of male infertility is attributed to the presence of OS in the reproduction system.”.
As minor comments, I draw attention to the frequent incorrect narrative and evaluation. For example:
Page 1 line 42. The cited work (3) does not cover all the aspects cited by the authors. It is better to use doi: 10.1023/a:1027304122259
Response: Thanks for the suggestion. The references have been added.
Page 2 line 52. OXPHOS does not produce ROS. ROS is produced in the respiratory chain (ETC)
Response: Thanks for the suggestion. We have revised the sentence as suggested.
“…Electron leakage in the electron transport chain (ETC) also produces reactive oxygen species (ROS) as a by-product...”
Page 3 line 90. What other organelles and cytoplasm are you talking about?
Response: Thanks for the suggestion. We have revised the sentence as suggested.
“…with the disappearance of other cytoplasm organelles, such as the endoplasmic reticulum or Golgi apparatus...”
Page 3 line 100. What isoenzymes are you talking about
Response: Thanks for the suggestion. Sorry for the typo, it should be “such as cytochrome oxidase, succinate dehydrogenase”. We have corrected the typo.
Page 3 line 111. Again, references (3, 17) do not correspond to the context and it is better to cite doi: 10.1023/a:1027304122259
Response: Thanks for the suggestion, the references have been added.
Page 3 line 113. The word “mostly” is wrong and should be replaced by “exclusively”.
Response: Thanks for the suggestion. We have revised the sentence as suggested.
“…ATP in sperm is exclusively produced via two metabolic pathways…”
Page 5 line 188. The phrase is incorrect, because oxygen consumption is a complex function of the membrane potential. High oxygen consumption can be caused by depolarization of the mitochondrial membrane, therefore the word “positively” is true for the membrane potential and incorrect for oxygen consumption.
Response: Thanks for the suggestion. We have revised the sentence as suggested.
“Numerous studies have confirmed that the MMP is positively correlated with sperm motility.”
Page 6. Lines 202-203. The phrase is not accurate, because a certain increase in free Ca in the matrix will activate three dehydrogenases, however, at higher values, nonspecific permeability may occur, which is fatal.
Response: Thanks for the suggestion. We have revised the sentence as suggested.
“Appropriate concentration of Ca2+ in the matrix can promote energy generation, thus increasing intracellular ATP content and improving sperm motility”
Page 7. Line 243. The phrase is not correct. If apoptosis has already occurred, then the cell is dead and there can be no question of any motility.
Response: Thanks for the suggestion. We have revised the sentence as suggested.
“…ultimately leading to decreased sperm count and infertility…”
Reviewer 5 Report
This review covers an important aspect, which is less studied and a huge clinical implication. In my opinion if the author also add current information about the following aspects, it will add more importance to this review.
The author said that in the initial stage of spermatogenesis, the mitochondria are flat, small, round or oval, and their number increases significantly. Therefore, mitochondrial fission and fusion may be important events in spermatogenesis and sperm maturation. What is the role of mitochondrial fission and fusion in spermatogenesis and sperm maturation?
Lipid is an important component for maintenance of sperm structure and function. Mitochondrial damage results in lipid peroxidation, alteration in lipid synthesis and lipogenesis. Author can add the available information about the mitochondria-mediated lipid changes in the sperm in the aspect of male infertility.
As per the title of the review, author wanted to study the information about “age-related decline of male fertility”. In the review, author successfully discussed about the mitochondrial damage and male infertility. However, age related changes was not discussed in equal attention. As per the title of the study, age related mitochondrial changes in sperm and its relationship with the infertility need to be discussed. .
Author discussed that some antioxidants did not show improvement of sperm quality after oral administration. Author may discuss about the absorbance of antioxidants in the tissue in this aspect. All the dietary and orally administered antioxidants do not get absorbed but the target tissue to the therapeutic levels. What are the reports about the absorbance of the antioxidants discussed here?
The grammar and language of the review are overall satisfactory, but there are some extra spaces between two words and sometime no space. Please edit those.
Author Response
We thank the reviewers for their insightful suggestions and comments on the manuscript.
This review covers an important aspect, which is less studied and a huge clinical implication. In my opinion if the author also add current information about the following aspects, it will add more importance to this review.
The author said that in the initial stage of spermatogenesis, the mitochondria are flat, small, round or oval, and their number increases significantly. Therefore, mitochondrial fission and fusion may be important events in spermatogenesis and sperm maturation. What is the role of mitochondrial fission and fusion in spermatogenesis and sperm maturation?
Response: Thanks for the suggestion. We have included the part in the revised manuscript.
“Thus, mitochondrial fusion and fission (mitochondrial dynamics) realizes the dynamic changes in mitochondrial number and structure during spermatogenesis, allowing fine-tuned responses to cellular energy demands[6,25]. In addition, through mitochondrial dynamics, mitochondria in mature sperm are highly concentrated, and the mitochondrial metabolism is more efficient at this time, which also plays a key role in maintaining sperm motility and fertilization[22,26]. However, the role of mitochondrial dynamics during spermatogenesis and sperm maturation remains largely unknown.”
Lipid is an important component for maintenance of sperm structure and function. Mitochondrial damage results in lipid peroxidation, alteration in lipid synthesis and lipogenesis. Author can add the available information about the mitochondria-mediated lipid changes in the sperm in the aspect of male infertility.
Response: Thanks for the suggestion. We have included the part in the revised manuscript.
“…Sperm cell membranes are rich in a variety of unsaturated fatty acids, which renders them more vulnerable to ROS-induced lipid peroxidation. Interestingly, OS-induced lipid peroxidation has also been shown in human sperm to generate electrophilic aldehydes, such as 4-hydroxynonenal and acrolein. These compounds react with mitochondria by targeting succinate dehydrogenase, leading to changes in respiratory chain function and activation of apoptotic pathways, all of which lead to overproduction of ROS [6].”
As per the title of the review, author wanted to study the information about “age-related decline of male fertility”. In the review, author successfully discussed about the mitochondrial damage and male infertility. However, age related changes was not discussed in equal attention. As per the title of the study, age related mitochondrial changes in sperm and its relationship with the infertility need to be discussed. .
Response: Thanks for the suggestion. We have included the parts in the revised manuscript.
“3.7 Age-related male infertility and the changes of sperm mitochondrial
“With the increase of male fertility age worldwide, more and more studies begin to focus on the impact of advanced paternal age (APA) on male fertility and offspring health. The most commonly used criterion to define advanced paternal age (APA) is age > 40 years at conception[11]. The aging of advanced male germ cells is multifaceted (Figure 3), including decreased semen volume, sperm count, motility and normal morphology, increased sperm DFI and methylation, all of which may lead to male infertility and adverse effects on offspring[56,57].
A systematic meta-analysis examining the effects of aging on semen parameters found that men over 50 years experienced a 3-22% decrease in semen volume, 3-37% in sperm motility, and a 4%-18% decrease in normal morphology compared with men under 30-years. Meanwhile, the pregnancy rate in the advanced age group was relatively reduced by 23-38%[58]. Another study in 2013 analyzed the semen data of 5081 men aged 16.5-72.3 years and found that the semen parameters did not change significantly before the age of 34, while the total sperm count, sperm motility, and normal morphology decreased year by year[59]. Studies also showed similar results that male aging was accompanied by a decrease in semen volume, total sperm count, sperm motility, normal-morphed sperm, and an increase in DFI[60]. A latest systematic review found that APA was associated with increased DFI, which suggested that DFI should be assessed in infertile older men with normal semen parameters to better identify the etiology of infertility[61]. Another study in 2013 analyzed sperm 5-methylcytosine (5-mC) and 5-hydroxymethylcytosine (5-hydroxymethylcytosine, 5-hmC) levels and found that sperm DNA methylation status was stable in the short term, but increased with age, with an average annual increase of 1.76% for 5-mC and nearly 5% for 5-hmC. The above studies, together,show that male fertility declines gradually with age[62].
Mitochondrial structure and function alter during aging. Aging can lead to a decrease in sperm MMP and cause oxidative damage to mtDNA[12,63]. Studies have also confirmed that the antioxidant capacity of aging male sperm was reduced, while the level of ROS and lipid peroxides were increased[11,64]. In addition, mitochondria in hamsters undergo significant morphological change with aging, including mitochondrial swelling, mitochondrial vacuolization, and significantly reduced cristae[65]. “
“4.3. Antioxidants in the treatment of aging infertile men
There are many factors that causing the decline of semen quality in aging male, one of which is the toxic effect of OS[96,97]. Mitochondria and sperm plasma membrane are the two major sites for ROS generation in sperm. With the increase of age, the human antioxidant capacity gradually declines, and the generated ROS cannot be removed effectively to induce OS, resulting in lipid peroxidation, DNA fragmentation, enzymatic denaturation and damage to sperm mitochondrial structure and function. The damaged mitochondria will produce more ROS and fall into a vicious circle, resulting in the decline of fertility in aging male[98,99]. Then, antioxidant therapy for aging male should be effective.
Indeed, there are only a small number of studies on antioxidant treatments specifically for aging male (table 2). Some studies have found that antioxidants can improve sperm quality in aging male[100,101], and some have found no improvement[100]. In addition, in vitro studies have found that adding idebenone (a mitochondrial-permeable synthetic benzoquinone that acts as an antioxidant by scavenging free electrons) to the sperm of men over 40 years old can reduce sperm ROS and improve fertilization rate[102]. However, the sample size of these studies is small, and the combination of multiple antioxidants is generally used, which cannot effectively prove the specific mechanism of antioxidants.”
Author discussed that some antioxidants did not show improvement of sperm quality after oral administration. Author may discuss about the absorbance of antioxidants in the tissue in this aspect. All the dietary and orally administered antioxidants do not get absorbed but the target tissue to the therapeutic levels. What are the reports about the absorbance of the antioxidants discussed here?
Response: Thanks for the suggestion. We have revised the manuscript as suggested. We included the discussion about the absorbance and the antioxidant effects in the target tissues.
“Some antioxidants did not show improvement of sperm quality after oral administration. This could be due to the inefficient absorbance of antioxidants in the tissue. Most of the antioxidants can cross plasma membrane and scavenge the toxic consequences of ROS in the cells, yet, it is important to test whether the dietary and orally administered antioxidants get absorbed by the target tissue to the therapeutic levels and to evaluate the effects. With the wide application of antioxidants, the determination methods of antioxidant effects and abilities are constantly developing and improved. Given that male infertile patients usually show higher levels of ROS and malondialdehyde (MDA), accompanied by a decrease in TAC, clinical evaluation of antioxidant effects is generally performed by chemiluminescence or colorimetry to detect seminal plasma ROS, MDA and TAC concentrations[2,75,76]. However, these methods all measure the anti-radical capacity of antioxidants in the final state of the reaction. Oxygen radical absorbance capacity (ORAC) is an exciting and revolutionary analytical method based on fluorescence decay, which can dynamically monitors the process of antioxidants inhibiting free radical chain reactions[103,104]. The advantages of this method are close to the body physiological conditions, complete chemical reaction, simple operation and high sensitivity, and not easy to be disturbed by human factors. The study in 2020 evaluated the redox status of non-obstructive azoospermia (NOA) patients through ORAC, and found that the production of ROS may be directly related to spermatogenesis disorders[105]. Unfortunately, no relevant studies of ORAC application to detect the effect of antioxidants in treating male infertility have been retrieved.”
The grammar and language of the review are overall satisfactory, but there are some extra spaces between two words and sometime no space. Please edit those.
Response:
Thanks for the suggestion. We have revised the manuscript as suggested.
Round 2
Reviewer 2 Report
Thank you very much for addressing all the raised points. Best wishes.
Author Response
We greatly appreciate the effort and time devoted by the reviewer while evaluating this manuscript! Thanks for the favorable comments!
Here we attached the revised manuscript.

Reviewer 4 Report
I think that the current version of the review sounds better than the original one and the authors properly addressed my comments. The only one note, please change the title of the paragraph 3.7 from Age-related male infertility and the changes of sperm mitochondrial to Age-related male infertility and the changes of sperm mitochondria.
Author Response
We greatly appreciate the effort and time devoted by the reviewer while evaluating this manuscript. Thanks for the favorable comments.
We have changed the title of the paragraph 3.7 from “Age-related male infertility and the changes of sperm mitochondrial” to “Age-related male infertility and the changes of sperm mitochondria.” Thanks again for the suggestion.
